# Sightless but Not Blind: A Non-Ideal Spectrum Sensing Algorithm Countering Intelligent Jamming for Wireless Communication

**Ziming Pu** [1], **Yingtao Niu** [1,*], **Peng Xiang** [2] and **Guoliang Zhang** [1]

1   The Sixty-Third Research Institute, National University of Defense Technology, Nanjing 210007, China
2   College of Communications Engineering, Army Engineering University of PLA, Nanjing 210007, China
*   Correspondence: niuyingtao78@hotmail.com

**Abstract:** Aiming at the existing intelligent anti-jamming communication methods that fail to consider the problem that sensing is inaccurate, this paper puts forward an intelligent anti-jamming method for wireless communication under non-ideal spectrum sensing (NISS). Under the malicious jamming environment, the wireless communication system uses Q-learning (QL) to learn the change law of jamming, and considers the false alarm and missed detection probability of jamming sensing, and selects the channel with long-term optimal reporting in each time slot for communication. The simulation results show that under linear sweep jamming and intelligent blocking jamming, the proposed algorithm converges faster than QL with the same decision accuracy. Compared with wide-band spectrum sensing (WBSS), an algorithm which failed to consider non-ideal spectrum sensing, the decision accuracy of the proposed algorithm is higher with the same convergence rate.

**Keywords:** non-ideal spectrum sensing; intelligent anti-jamming; Q-learning; wide-band spectrum sensing

## 1. Introduction

In the past 20 years, wireless communication technology was widely used. However, due to the openness of the wireless channel, the challenge of artificial interference in wireless communication is becoming more and more serious. Artificial interference mainly includes two types. One is unintentional interference [1], and the other one is intentional jamming. Intentional jamming refers to the jamming behavior taken for the purpose of destroying the information transmission process of a wireless communication system. According to whether the strategy is fixed or not, artificial intentional jamming can be divided into fixed-strategy jamming and dynamic strategy jamming. Fixed-strategy jamming mainly includes multi-tone jamming, partial-band jamming, periodic-pulse jamming, linear-sweep jamming, etc., and its strategy is fixed and the law is easily perceived. Dynamic strategy jamming mainly includes dynamic probability jamming, intelligent blocking jamming, etc., and its jamming law is not easily obtained through simple observation or sensing.

### 1.1. Related Works

In recent years, the continuous development of machine learning algorithms provided new intelligent ideas for communication anti-jamming. In the frequency domain, the author in [2] modeled the problem of multi-channel jamming and anti-jamming as a Markov decision process (MDP). The best defense strategy is obtained through value iteration under the channel transition probability, and rewards are completely known. Similarly, in [3], the author modeled the game problem between the secondary user and the jammer in the cognitive radio system as MDP, using Q-learning and maximum likelihood estimation to obtain attacker parameters and obtain the optimal channel switching strategy. In [4], the author also modeled the jamming and anti-jamming process as MDP, and proposed a game

theory anti-jamming scheme (GTAS), which achieved higher returns. In [5,6], the authors also studied the anti-jamming problem of user channel selection by using the method of game theory. In [7], the author used Q-learning to solve the channel allocation problem, and proposed a channel allocation strategy with the lowest failure rate. In [8], the author used MDP to model dynamic and complex spectrum environment, and used Q-learning to obtain the optimal communication strategy. Because Q-learning has the problem of disaster maintenance, deep learning is introduced to solve the complex anti-jamming strategy learning problem. In [9], the author proposed a hierarchical deep reinforcement learning algorithm without a jamming mode and channel model, which solved the problem of selecting many optional frequencies in the jamming environment. In the power domain, the author in [10] proposed a power control strategy based on reinforcement learning for the case of unknown jamming patterns and channel parameters, which improved the communication efficiency. Aiming at multi-domain joint jamming in the frequency domain, time domain, and power domain, the authors in [11] proposed a multi-parameter intelligent anti-jamming method based on 3D Q-learning. The proposed algorithm has a lower jamming collision rate than the traditional Q-learning algorithm. The authors in [12] proposed the CAAQ algorithm to solve the problem of multi-user cooperative anti-jamming. By increasing the distance threshold, the problem of mutual jamming between users was well avoided. The author in [13] proposed an anti-jamming power control strategy based on Q-learning, which has better performance in the condition of the game model unknown. However, the above mentioned algorithms require the acknowledgment character (ACK) fed back from the receiver after data packets are successfully received as a means to sense whether the communication channel is subjected to jamming, which increases the operation overhead and can induce additional risk of ACK frame jamming. Furthermore, those strategies can only update the Q value of the selected channel one by one, and their convergence rates are low.

To address these issues, the author in [14] used the wide-band spectrum sensing technology to sense the jamming state of multiple channels simultaneously, which significantly improved the convergence rate of the algorithms without the need of an ACK feedback channel. However, the strategy proposed in [14] assumed that the observed jamming state by the system was the actual jamming state of the system, which neglected the non-ideality of the jamming state observation. In practice, due to the existence of many non-ideality factors, such as false alarm and the missed jamming states in the observed results, the observed jamming state is not necessarily equivalent to the actual jamming state of the system, which may lead to large uncertainties in the algorithm. Therefore, if we ignore these undesirable characteristics of perception, we will not be able to correctly learn the behavior of jamming, which will lead to large disturbances in the algorithm. Aiming at the problem that channel quality varies with probability, reference [15] studied the time-constrained downlink scheduling strategy for the actual channel observation environment, proposed a simplified partially observable Markov decision process (POMDP) modeling method for downlink transmission, and proposed a low-complexity suboptimal strategy method based on finite time domain Q_MDP algorithm.

Inspired by the work in [14], this paper proposes an intelligent anti-jamming method for wireless communication in the case of non-ideal spectrum sensing. Compared with the traditional Q-learning algorithms in Refs. [3–13], this paper uses the idea of broadband sensing technology to update the Q value of multiple channels at a time; compared with WBSS algorithms in Ref. [14], this paper considers the non-ideal sensing, which is closer to the actual electromagnetic environment. Firstly, the algorithm models the problem of communication channel selection in a jamming environment with false alarm and missed detection as an improved Markov decision process (IMDP) combined with false alarm and missed detection. Then, it is solved by the NISS algorithm, obtaining the selection of the optimal communication channel. The advantages for decision accuracy and convergence rate of the proposed algorithm are proved comparison to the traditional Q-learning [16]

and the conventional wide-band spectrum sensing algorithm [14] through the MATLAB simulation results.

### 1.2. Contribution and Structure

The contribution of this paper is as follows:

- This paper proposes a NISS algorithm, which combines the advantages of Q-learning and the WBSS algorithm. The proposed algorithm has a fast convergence rate and high decision accuracy.
- This paper takes the probability of false alarm and missed detection into account in anti-jamming communication for the first time, which is closer to the actual electromagnetic environment and fills the blank of intelligent anti-jamming wireless communication in the case of non-ideal sensing.

The remainder of this paper is organized as follows. Section 2 presents the system model and problem formulation. In Section 3, we introduce the detailed derivation of the NISS algorithm. The simulation results and analysis are presented in Section 4. Our concluding remarks are given in Section 5.

## 2. System Model and Problem Formulation

### 2.1. System Model

Figure 1 shows the model of the communication system. There is one communication transmitter, one communication receiver, and one jammer. The communication transmitter and receiver can use the communication channel to communicate. There are $M$ possible channels in the wireless communication system, which are recorded as $\{1, 2, 3, \ldots, M\}$, selecting one channel at a time for communication. The jammer has a non-intelligent mode and an intelligent mode. In the non-intelligent mode, the jammer performs conventional jamming, such as linear sweep; in the intelligent mode, the jammer has a greater probability of jamming with high-communication frequency channels according to the communication frequency of each channel detected in advance.

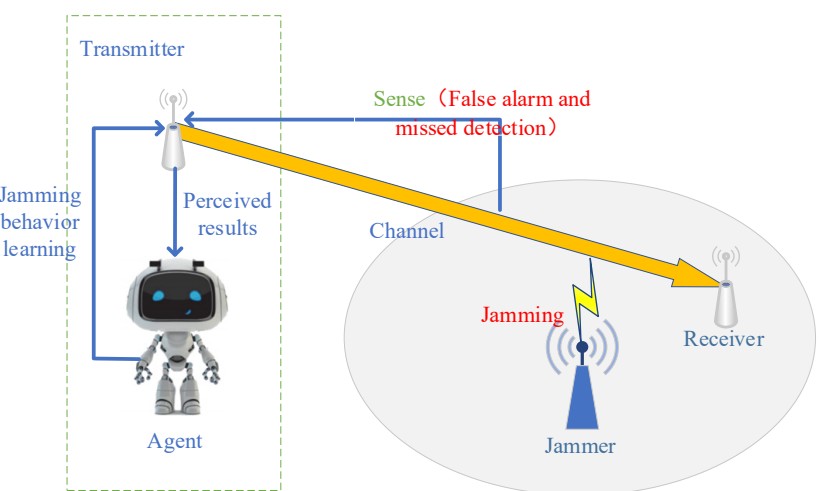

**Figure 1.** Communication system model.

Figure 2 shows the structure of communication time slot. Each jamming time slot corresponds to a communication time slot, which can be divided into transmission sub-slot, sensing sub-slot and learning sub-slot. In the transmission sub-slot, the transmitter selects an undisturbed channel to transmit information to the receiver according to the judgment of the previous time slot on the jamming channel. In the sensing sub-slot, the receiver senses each channel, and transmits the sensing results to the agent for learning. The agent obtains the judgment of the available channels of the same time slot in the next jamming

period. The transmitter selects the optimal channel for the communication transmission of the next time slot according to the judgment.

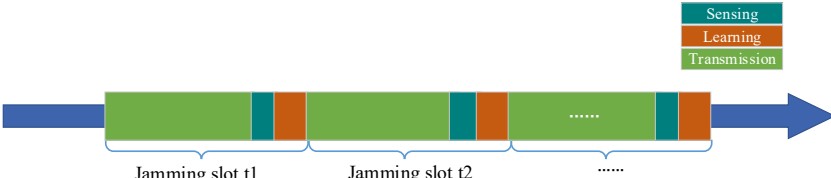

**Figure 2.** Structure of the communication time slot.

Figure 3 is a state transition diagram between the actual state and the observed (sensing) state for channel *i* due to the existence of false alarm and missed detection. Where $p_i$ is the missed detection probability, indicating that the observation state is no jamming, while the actual state is jamming. $q_i$ is the false alarm probability, indicating that the observation state is jamming, while the actual state is no jamming.

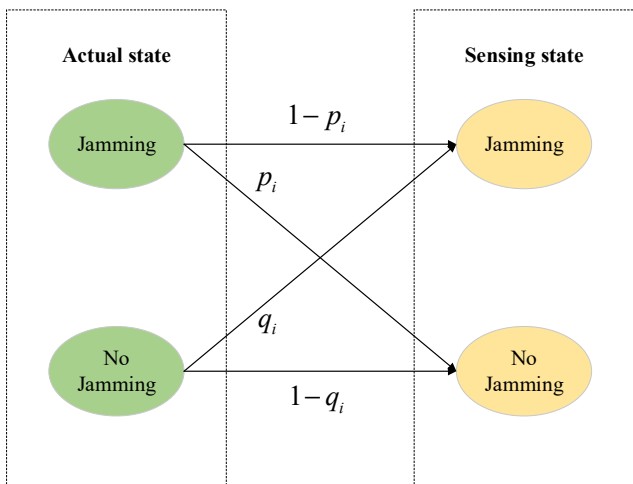

**Figure 3.** Transition diagram between actual state and observation state caused by false alarm and missed detection.

*2.2. Problem Formulation*

To simplify the research, the following assumptions are made:

- The communication frequency band is divided into *N* channels with the same bandwidth, and there is no frequency overlap between the channels, and the fading characteristics of each channel are the same and flat fading.
- The sensing result is only affected by false alarm and missed detection, which leads to inaccuracy, and there is no inaccuracy caused by other factors.
- In the same time slot, the channel of jamming does not change.

Based on the above assumptions, since the agent cannot accurately perceive the state of the system, we use the improved Markov decision process (IMDP) for modeling and solving. IMDP can be expressed as a five tuple $\langle S, A, P, O, r \rangle$, in which, in addition to the state space *S*, action space *A*, state transition probability *P*, and real-time reward function *r* of the general MDP, there is also observation space *O*.

The state space *S* can be expressed as:

$$S \triangleq \{n_1, n_2, \ldots, n_i : n_k \in \{1, 2, 3, \ldots, N\}\} \tag{1}$$

where $n_k = j$ indicates that the $k_{th}$ channel to be interfered is channel *j*, and *N* channels can be interfered at most in the same time slot.

Action space $A$ can be expressed as:

$$A \triangleq \{a : a \in \{1, 2, 3, \ldots, N\}\} \tag{2}$$

where $a = i$ indicates that the transmitter chooses to communicate on channel $i$.

Observation space $O$ can be expressed as:

$$O \triangleq \left\{ o_t = \left( o_t^1, o_t^2, \ldots, o_t^M \right) : o_t^i \in \{0, i\} \right\} \tag{3}$$

where $o_t^i = 0$ and $o_t^i = i$, respectively, indicate that the observation status of channel $i$ is undisturbed and disturbed in time slot $t$.

The reward function $r$ is defined as follows:

$$r_t^i = \begin{cases} 0 & a \neq i; \\ E & a = i \& i \notin s_t \\ -L & a = i \& i \in s_t \end{cases} \tag{4}$$

where $r_t^i$ represents the reward function of channel $i$ in time slot $t$, and $-L$ represents the loss in case of message transmission failure. $E$ represents the return of successful message transmission. $s_t$ represents the jamming status of each channel at time $t$, and $s_t \in S$.

The observed state and actual state transition diagram of each time slot are shown in Figure 4.

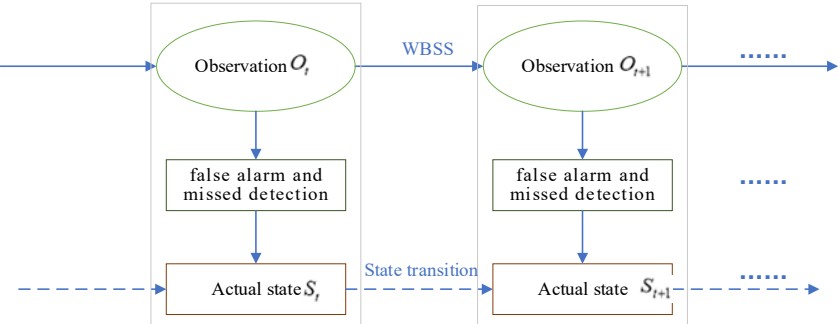

**Figure 4.** The observed state and actual state transition diagram of each time slot.

Where the observed states in time slot $t$ and time slot $t + 1$ can be obtained through wide-band spectrum sensing. At the same time, the false alarm probability $P_f$ and missed detection probability $P_d$ can be calculated according to the energy detection theory [17].

$$P_d = \mathrm{Pr}(Y > \lambda | H_1) = Q_u \left( \sqrt{2\gamma}, \sqrt{\lambda} \right) \tag{5}$$

$$P_f = \mathrm{Pr}(Y > \lambda | H_0) = \frac{\Gamma(u, \lambda/2)}{\Gamma(u)} \tag{6}$$

where $u = TW$ is time domain bandwidth product, and $\gamma$ is jamming to noise ratio (JNR), and $\lambda$ is decision threshold of energy detection.

### 3. Detailed Derivation of Algorithm

Q-learning is a form of typical model-free learning. Its basic idea is to establish a Q table. The values in the table represent the long-term cumulative rewards of executing the current strategy after the state $s_t$ selects the action $a_t$. The long-term cumulative reward can be expressed as follows:

$$V = E \left[ \sum_{\tau=0}^{+\infty} \gamma^\tau r_{t+\tau} \right] \tag{7}$$

where $\gamma$ is the discount factor, indicating the importance of future returns, and $r_t$ is the immediate reward value obtained in step $t$. The goal of Q learning is to find a strategy $\pi$ to maximize the long-term cumulative rewards under this strategy.

To solve the optimal strategy, the state value function $V$ and the state action value function $Q$ are defined as follows:

$$V^{\pi}(s) = E_{\pi}\left[\sum_{\tau=0}^{+\infty}\gamma^{\tau}r_{t+\tau}|s_t = s\right] \tag{8}$$

$$Q^{\pi}(s,a) = E_{\pi}\left[\sum_{\tau=0}^{+\infty}\gamma^{\tau}r_{t+\tau}|s_t = s, a_t = a\right]. \tag{9}$$

Since the MDP model is satisfied, it can be converted into a recursive form as follows:

$$\begin{aligned}V^{\pi}(s) \quad &= E_{\pi}\{r_t + \gamma[r_{t+1} + \gamma(r_{t+2} + \dots)]|s_t = s\}\\ &= \sum_{a\in A}\pi(a|s)\sum_{s'\in S}\{P(s'|s,a)[r(s'|s,a) + \gamma V^{\pi}(s')]\}\end{aligned} \tag{10}$$

$$Q^{\pi}(s,a) = \sum_{s'\in S}\{P(s'|s,a)[r(s'|s,a) + \gamma V^{\pi}(s')]\} \tag{11}$$

where $P(s'|s,a)$ represents the probability of taking action $a$ represents the probability of taking action in state $s$ and transferring the state to $s'$, and $r(s'|s,a)$ represents the corresponding reward.

According to the Bellman optimization principle, the optimal value $Q^{\pi^*}(s,a)$ can be obtained as follows [18]:

$$Q_*^{\pi}(s,a) = \sum_{s'\in S}\left\{P(s'|s,a)\left[r(s'|s,a) + \gamma\max_{\pi}Q^{\pi}(s,a)\right]\right\} \tag{12}$$

$$Q^{\pi^*}(s,a) = \max_{\pi}Q_*^{\pi}(s,a). \tag{13}$$

Therefore, the optimal strategy $\pi*$ can be obtained as follows:

$$\pi* = \arg\max_{\pi}\{Q^{\pi}(s,a)\} \tag{14}$$

Since the Q-learning algorithm does not need prior knowledge, such as state transition probability, its update formula is as follows:

$$Q_{t+1}(s,a) = (1-\alpha)Q_t(s,a) + \alpha\left(r_t + \gamma\max_{a'}Q_t(s,a')\right) \tag{15}$$

where $\alpha$ is the learning rate. The reference [19] and [20] proved that if $\alpha$ meets the conditions:

$$\alpha_t \in [0,1), \sum_t^{\infty}\alpha_t = \infty, \text{ and } \sum_t^{\infty}(\alpha_t)^2 < \infty \tag{16}$$

then the Q-learning algorithm can converge after finite iterations. When the Q table converges, the action corresponding to the maximum Q value in each state is the optimal action in that state.

Wide-band spectrum sensing algorithm (WBSS) senses multiple channels in the same time slot, obtains the actual state of each channel (whether it is jammed) according to the sensing, and updates the Q value of each channel at the same time slot. Therefore, compared with the conventional Q-learning algorithm [16], which only updates the Q value of the selected channel in one time slot, the convergence rate of the WBSS algorithm will be greatly improved.

The Q value of the WBSS algorithm is updated as follows:

$$Q_{t+1}(s,n_i) = (1-\alpha)Q_t(s,n_i) + \alpha\left[r_t + \gamma\max_{n_j}Q_t(s,n_j)\right] \tag{17}$$

where $n_i$ refers to different channels, and its value is $\{1, 2, 3, \ldots, M\}$. That is, for any time slot, the Q values of the $M$ channels are updated at the same time. $r_t$ represents the instant benefit of the time slot $t + 1$ selecting channel $n_i$ for communication.

It can be seen from Equations (11) and (12) that to update the Q value and obtain the optimal strategy, it is necessary to know the state of the current time slot. However, in the actual electromagnetic environment, due to the inaccuracy of observation, the state of the current time slot is not a completely determined state, but there are multiple possible states related to the probability of false alarm and missed detection. Therefore, different from the conventional WBSS algorithm, the proposed algorithm takes the false alarm and missed detection probability into account when calculating the Q value and making decisions, obtaining the NISS algorithm.

Since there are different states, such as $s_t^1 = \{1, 0, 0, \ldots, 0\}$ (only channel 1 is jammed) and $s_t^2 = \{1, 2, 0, \ldots, 0\}$ (only channel 1 and channel 2 are jammed) for actions such as $a = 1$ (select channel 1 for communication), the communication results are the same and the benefits are the same. Therefore, we change the state from the set of all channel states as one state to the time slot as the state. When calculating the immediate return $r$, we only need to consider the actual state of the selected communication channel, not the actual state of other channels.

Then, for the time slots $n_{t-1}$ and $n_t$, where the system is located, $o_{t-1}, o_t \in \mathbf{O}$ is observed. For each $a_t \in \mathbf{A}$, the Q value is calculated as follows:

If the observation of the selected channel is jamming, that is $a_t \in o_{t-1}$, update the Q value as Equation (18).

$$Q(n_{t-1}, a_t) = Q(n_{t-1}, a_t) + \alpha \left[ p_i r_1 + (1 - p_i) r_2 + \gamma \max_{a_{t+1} \in A} Q(n_t, a_{t+1}) - Q(n_{t-1}, a_t) \right] \quad (18)$$

If the observation of the selected channel is no jamming, that is $a_t \notin o_{t-1}$, update the Q value as Equation (19).

$$Q(n_{t-1}, a_t) = Q(n_{t-1}, a_t) + \alpha \left[ (1 - q_i) r_1 + q_i r_2 + \gamma \max_{a_{t+1} \in A} Q(n_t, a_{t+1}) - Q(n_{t-1}, a_t) \right] \quad (19)$$

For the time slot $n_t$, the observation is $o_t$. We obtain the actions as Equation (20).

$$a_{t+1} = \arg \max_{a_{t+1} \in A} Q(n_t, a_{t+1}) \quad (20)$$

Algorithm 1 is the flow of intelligent anti-jamming communication decision algorithm based on NISS.

---

**Algorithm 1:** Intelligent anti-jamming communication decision algorithm based on NISS.

---

1. **Initialization:** Learning factor $\alpha$, Discount factor $\gamma$ and other parameters in Table 1.
The Q table is initialized as a zero matrix with $N_T$ rows and $M$ columns, that is, for any $n_T$ and $a$, let $Q(n_T, a) = 0$.
2. **for** $t = 1, 2, \ldots T$ **do**
3. In the current transmitter state $s_t$, the transmitter performs the optimal policy selection action $a_t$ obtained in the last timeslot or the initial action $a_0$.
4. The transmitter detects the energy of each channel.
5. Calculate the probability of false alarm $p_f$ and missed detection $P_m$ according to the detection results.
6. According to the detection results, false alarm, and missed detection, the real-time reward $r$ is calculated and the next state $s_{t+1}$ is predicted to obtain the optimal communication channel $a_{t+1}$.
7. The agent updates the Q value according to (18) and (19).
8. The agent obtains the optimal strategy $\pi*$ according to (20) and instructs the transmitter to transmit in the next time slot.
9. $t = t + 1$
10: **end for**

---

First, initialize the system. Second, according to the last decision result, the optimal communication channel is selected. Third, the transmitter detects the energy of all channels. Fourthly, the false alarm probability and missed detection probability are calculated according to the energy detection results, the reward of each channel is obtained, and the Q table is updated. Finally, the optimal communication channel of the next slot is selected according to the Q table, and an iteration is completed.

## 4. Simulation Result and Analysis

### 4.1. Parameter Settings

Table 1 shows the simulation parameters.

**Table 1.** Simulation parameter settings.

| Parameters | Value |
| --- | --- |
| Communication timeslot length $T_s$ | 0.6 ms |
| Transmission timeslot length $T_{trans}$ | 0.5 ms |
| Perception timeslot length $T_{sensing}$ | 0.04 ms |
| Learning timeslot length $T_{learning}$ | 0.06 ms |
| Total transmission timeslots $T_a$ | 10,000 |
| Number of available channels $N$ | 10 |
| Transmission power of transmitter $P$ | 30 dBm |
| Fading of communication signal $B_S$ | $-130$ dB |
| Transmission power of jamming $J$ | 30 dBm |
| Fading of jamming $B_J$ | $-134$ dB |
| Power spectral density of ambient noise | $-174$ dBm/Hz |
| Channel bandwidth $B_W$ | 1 MHz |
| Learning rate factor $\alpha_m$ | 0.1 |
| The discount factor $\gamma$ | 0.5 |
| Transmission success reward $E$ | 1 |
| Transmission failure loss $L$ | $-3$ |

According to the parameters in Table 1, the jamming noise ratio (JNR) can be calculated as 10 dB, and the time bandwidth product is 5, so we can calculate the false alarm probability as $P_f = 0.0549$ and missed detection probability as $P_m = 0.1021$ according to Equations (5) and (6). To evaluate the performance of this algorithm, this algorithm is compared with the traditional QL algorithm and WBSS algorithm.

In this paper, the effectiveness and universality of the algorithm will be verified by simulation from both fixed-strategy jamming and dynamic strategy jamming. The first is the fixed-strategy jamming, and the linear sweep jamming is selected as the research object. Figure 5 shows the time-frequency distribution of linear sweep jamming, and the red background indicates jamming. The jamming channel changes linearly at any time slot, and 10 time slots are a jamming period. In the same time slot, the jamming channel does not change.

The second is the dynamic strategy jamming, and the intelligent blocking jamming is selected as the research object. Intelligent blocking jamming refers to a jamming strategy in which the jammer selects the channels with the highest number of communications to interfere with the relative number of communications in each channel in the previous period of time.

Figure 6 shows the probability distribution of intelligent blocking jamming. The number represents the jamming probability of the channel in that time slot, which is determined by the proportion of the communication times of each channel in the total communication time of the previous period. A jamming period is 10 time slots.

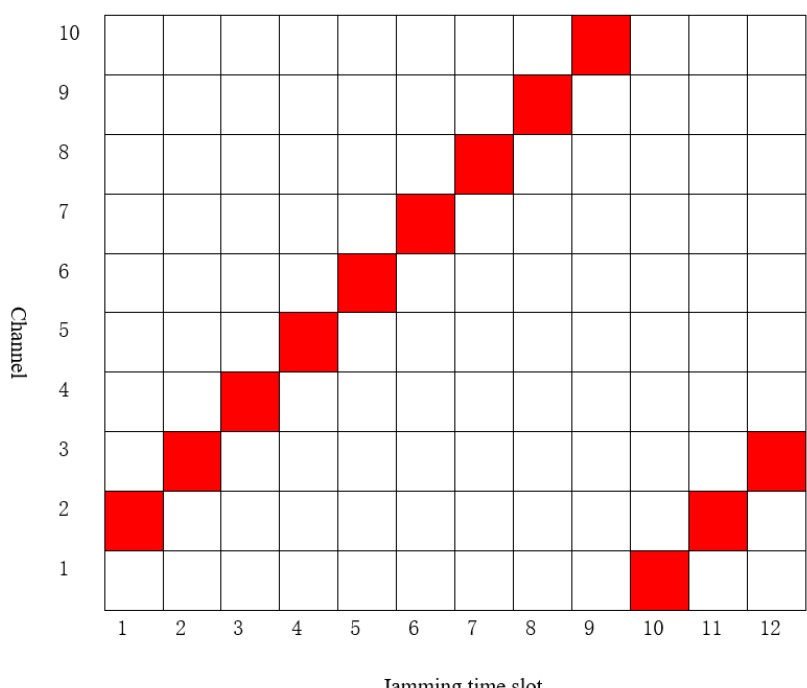

**Figure 5.** Linear sweep jamming distribution.

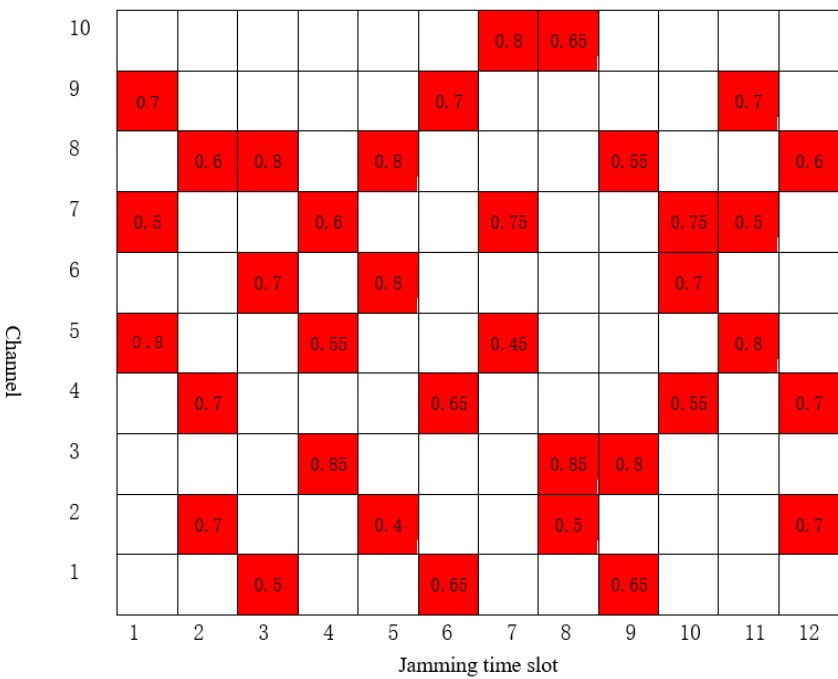

**Figure 6.** Probability distribution of intelligent blocking jamming.

### 4.2. Result Analysis

Figure 7 is a comparison of the decision accuracy (ratio of successful transmission times to total communication time) of the proposed algorithm anti-linear sweep jamming with traditional Q-learning and wide-band spectrum sensing algorithms. The decision accuracy of traditional Q-learning converges to 100% after about 30 rounds of algorithm iteration. To accelerate the convergence of the algorithm, aiming at the problem that Q-learning only updates one Q value of the state action pair at a time, the WBSS algorithm senses the jamming states of all channels in each time slot, and updates the Q value of all

actions at the same time in each state, which greatly accelerates the convergence rate of the algorithm. However, due to false alarm $P_f = 0.0549$ and missed detection $P_m = 0.1021$, the sensing results cannot be completely accurate. For example, in a certain time slot, the channel state perceived is free of jamming, but the sensing result may be caused by missed detection, and the actual channel may be jammed. Therefore, the WBSS algorithm takes the sensing result directly as the actual state of the system and does not consider the impact of false alarm and missed detection on the sensing result. Its accuracy after convergence is only 90%, which is lower than the Q-learning algorithm. By taking the false alarm and missed detection probability into account, the inaccuracy of the sensing results and the decision of the optimal channel is more accurate and reasonable. Therefore, the decision accuracy of the NISS algorithm for the channel of linear sweep jamming can also reach 100%.

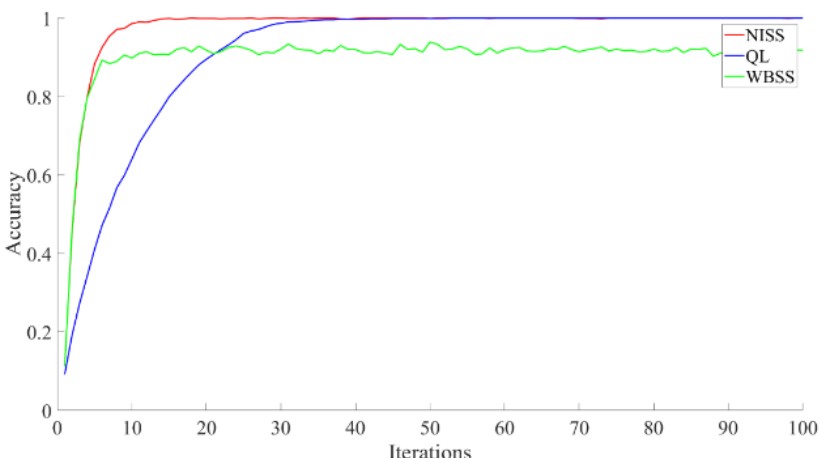

**Figure 7.** Decision accuracy of three anti-linear sweep jamming algorithms.

Since the actual jamming cannot be such regular linear sweep jamming, to verify the applicability of the algorithm under complex jamming patterns, the simulation verification against intelligent blocking jamming is carried out, and the results are shown in Figure 8.

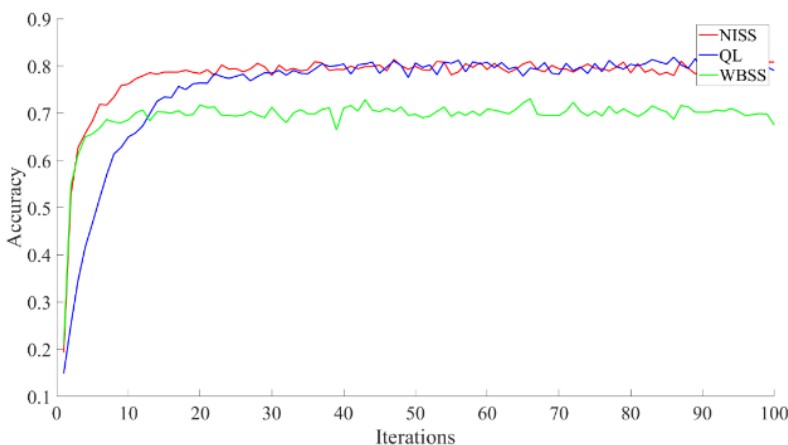

**Figure 8.** Decision accuracy of three anti-intelligent blocking jamming algorithms.

As can be seen from Figure 8, since the intelligent blocking jamming interferes with different channels in each time slot with probability, it is impossible to fully predict the jamming channel of the next time slot. Therefore, even with Q-learning, the decision accuracy is only 80%.

From the results of Figures 7 and 8, we can see that the NISS algorithm can converge faster than Q-learning with the same decision accuracy, and has higher decision accuracy than the WBSS algorithm with the same convergence rate.

Figure 9 is a comparison diagram of the change rule of successful transfer rate with JNR when the false alarm probability of the NISS algorithm and the WBSS algorithm is $P_f = 0.0549$. It can be seen from Figure 9 that with the increase in JNR, the decision accuracy of both algorithms increases. When JNR is taken from 2dB to 12dB, the decision accuracy of the NISS algorithm is significantly higher than that of the WBSS algorithm, which proves that the performance of the NISS algorithm is better than that of the WBSS algorithm. It should be noted that when JNR is low, the reason why the decision accuracy of the two algorithms is similar is that it is difficult to distinguish whether the channel contains jamming due to the low JNR. When JNR is very high, both decision accuracies reach the same maximum. The reason is that the energy of the jamming signal is very strong, the probability of missed detection is almost negligible, and the observation accuracy is only related to the probability of false alarm. The probability of false alarm is equal, so the decision accuracy is equal. The results show that the performance of NISS is greater than WBSS at the medium JNR.

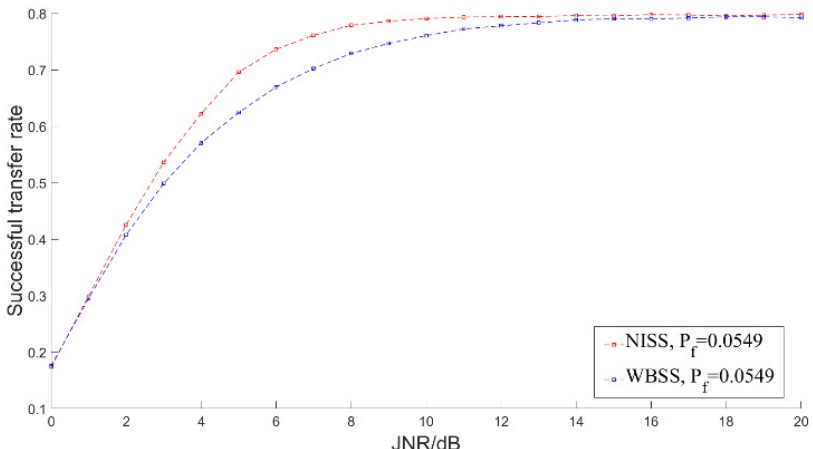

**Figure 9.** Variation diagram of decision accuracy with JNR of two anti-intelligent blocking algorithms.

Figure 10 is a channel decision diagram of the WBSS algorithm against intelligent blocking jamming. The vertical coordinate indicates the channel serial number, the horizontal coordinate indicates the current communication slot. The green area indicates that the current channel is predicted to be jammed, but there is no jamming in the actual channel. The light red area indicates that the current channel is jammed but was not predicted. The dark red area indicates that the current channel has jamming and was successfully predicted. For the WBSS algorithm, when the algorithm converges, some channels will be jammed, but the agent was not successfully predicted. If the channel is selected for communication at this time, the communication will fail and cause great losses.

Figure 11 is a channel decision diagram of the proposed algorithm against intelligent blocking jamming. Compared with Figure 10, after the algorithm converges, the NISS algorithm in this paper can accurately judge those channels that are actually jammed, and there is no missed detection. Although there will be no jamming but predicted jamming, the loss caused by this false alarm is very small. Therefore, the algorithm proposed in this paper can effectively solve the problem of wireless communication intelligent anti-jamming in the case of non-ideal spectrum sensing.

Comparing the performance of the algorithm in the anti-jamming channel decision under the two jamming patterns, we can see that: Although the accuracy of the Q-learning algorithm is high, the convergence rate of the algorithm is slow. For jammers with dynamic jamming patterns, they may not be able to learn the jamming rules in a short time and make effective anti-jamming decisions. In the anti-jamming channel decision of the WBSS

algorithm, although the convergence rate of the algorithm is much higher than that of Q-learning, there is a major defect in the sensing algorithm, that is, due to the problems of false alarm and missed detection, the sensing results are not necessarily accurate, so the accuracy of the jamming channel decision after convergence is low. The NISS algorithm is improved on the WBSS algorithm. By taking the sensing inaccuracy caused by false alarm and missed detection probability into account and the confidence of the actual channel state, it more accurately describes the current channel state in the time slot. Therefore, the NISS algorithm has the same convergence rate as the WBSS algorithm and does not lose the accuracy of the jamming channel decision.

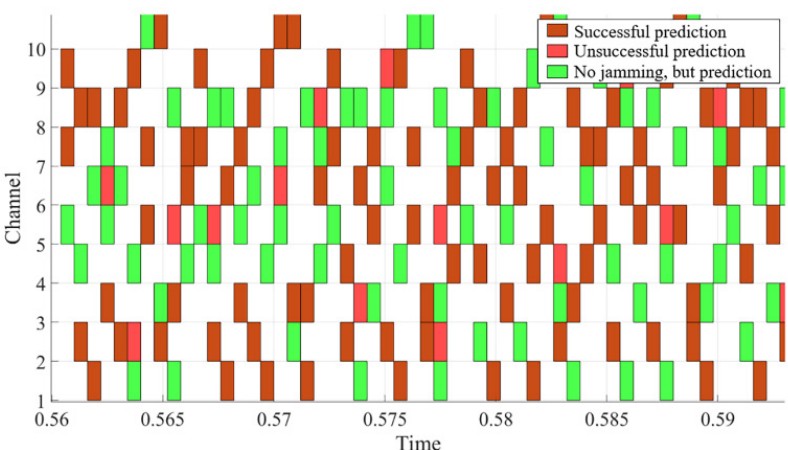

**Figure 10.** WBSS algorithm anti-intelligent blocking jamming channel decision diagram.

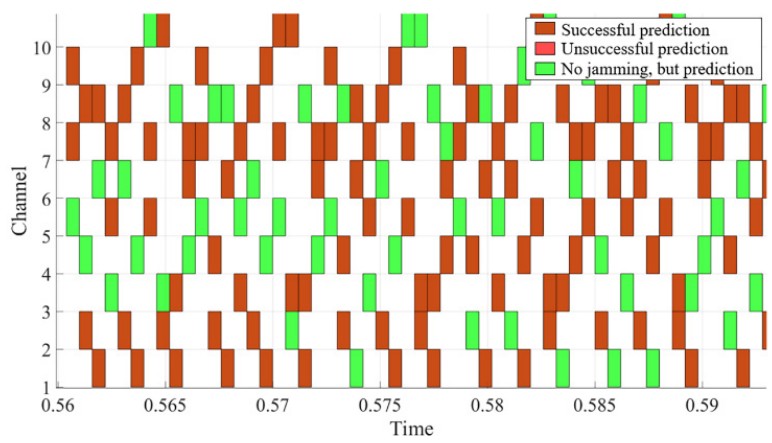

**Figure 11.** NISS algorithm anti-intelligent blocking jamming channel decision diagram.

## 5. Conclusions

This paper proposes a NISS intelligent anti-jamming algorithm. The main purpose of the proposed algorithm is to solve two problems. One is the problem of low convergence rate of Q-learning because of updating the Q value of each channel one-by-one, and the other problem is the non-ideal perception of the WBSS algorithm. By referring to the Q value update strategy of the WBSS algorithm and taking the probability of false alarm and missed detection into the calculation of the Q value, the proposed algorithm achieves good anti-jamming effect. The simulation is carried out under the conditions of linear sweep jamming and intelligent blocking jamming. The results show that compared with the traditional Q-learning algorithm, the proposed algorithm converges faster with the same decision accuracy; compared with the WBSS algorithm, when the convergence rate is the same, the accuracy of jamming channel decision making is higher, which fully shows that

this algorithm has better anti-jamming performance in the face of complex and changeable intelligent jamming.

**Author Contributions:** Methodology, Z.P. and Y.N.; writing—original draft, Z.P. and Y.N.; software, Z.P. and Y.N.; supervision, G.Z.; writing—review and editing, Y.N. and P.X.; validation, Z.P. and Y.N.; funding acquisition, Y.N.; project administration, Y.N. All authors have read and agreed to the published version of the manuscript.

**Funding:** This research was funded by the National Science Foundation of China (NSFC grant: U19B2014).

**Data Availability Statement:** Not applicable.

**Conflicts of Interest:** The authors declare no conflict of interest.

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
