# Peer review of "Sightless but Not Blind: A Non-Ideal Spectrum Sensing Algorithm Countering Intelligent Jamming for Wireless Communication"

_electronics, doi:10.3390/electronics11203402_

Round 1

Reviewer 1 Report

1. The motivation of this paper is not presented clearly. The authors should review related works more clearly and show main difference between this paper and the existing ones.

2. Almost references in this paper have been published long time ago. Again, the authors should update the related works with new references.

3. The main contribution of this paper in Sub-section 1.2  is not clear, which makes the Reviewer to evaluate the new points of this paper.

4.  Also, it is difficult to evaluate main contribution of proposing the new method as well as analyzing the performance in this paper.

5. The authors should compare the performance of the proposed method with new existing ones

6. The authors should add more explanation for Algorithm 1

7. This paper is not well-prepared. The authors should correct all the errors in this paper

- At page 1 of 12, reference [1] is cited not appropriately (please cite it similarly to other references). Similar to reference [14] at page 5 of 12.

- Please only use “Where” or “where” in this paper

- Looking at equation (10): size of texts in this equation is smaller than the other ones in this paper. Please correct the similar errors and strictly follow the journal template.

- Sentence “Figure 6. Probability distribution of intelligent blocking jamming” should be placed at the same page with the Figure 6.

- Figure 9 is not clear enough to read in the PDF format.

Author Response

Response to reviewer 1.

Reviewer 2 Report

The paper proposes an intelligent anti-jamming method for wireless communication under non-ideal spectrum sensing. the paper is well written and correctly structured.

I do have concern about the similarity of this paper with two other works, from the same author(s), recently published in a IEEE conference, that are not part of the reference list:

G. Zhang, Y. Li, L. Jia, Y. Niu, Q. Zhou and Z. Pu, "Collaborative Anti-jamming Algorithm Based on Q-learning in Wireless Communication Network," 2022 IEEE 2nd International Conference on Computer Communication and Artificial Intelligence (CCAI), 2022, pp. 222-226, doi: 10.1109/CCAI55564.2022.9807740.

Z. Pu, Y. Niu and G. Zhang, "A Multi-Parameter Intelligent Communication Anti-Jamming Method Based on Three-Dimensional Q-Learning," 2022 IEEE 2nd International Conference on Computer Communication and Artificial Intelligence (CCAI), 2022, pp. 205-210, doi: 10.1109/CCAI55564.2022.9807745.

In my opinion, the authors should include these two references in their work, and clearly specify the main differences between the proposed work and the previously published work. If the proposed work is a follow-up, the novelty should be also clearly specified. 

Author Response

Response to reviewer 2.

Reviewer 3 Report

The authors of present work paper present an intelligent anti-jamming method for wireless communication under non-ideal spectrum sensing, demonstrating through simulations that under the malicious jamming environment the proposed algorithm converges faster than QL, and the decision accuracy is higher compared with WBSS.

The presentation is fine, but some minor changes can improve the quality of the article:

Line 25-26 – The wide acceptance for "jamming" definition is as an intentional attack, while unintentional form of disruption is called "interference". The entire classification should take into consideration that jamming is considered by many as being only intentional.

Line 100 – Instead of "optional", wouldn't the word "possible" be more appropriate?

Lines 105-107 are a bit confusing. Maybe you should talk here about sub-slots: transmission sub-slot, sensing sub-slot and learning sub-slot.

In order to improve the readability of the article, the punctuation and the formatting of the text must be corrected in some parts of the article (there are a lot of full stops instead of commas, or situations such as line 196 that follows line 195 – where there's no need to start with capital letter). Also, after the line 226, there must be a clear indication of what's following after the colon (some bullets or numbers): at line 227 (the selected channel is jammed) and at line 230 (the selected channel is not jammed). Also, some equations have smaller fonts than others. For example, at line 229, the eq. 18 is barely distinguishable.

At lines 317-318, on the horizontal axis of Figure 10, it is the time represented actually, not the current communication slot. Also, wouldn't be better to have green color for successfully predictions and light/dark red for wrongfully predictions?

Author Response

Response to reviewer 3.

Reviewer 4 Report

The singular task has been reported in this paper called NISS for intellgent anti jamming mehtod. 

1.  The abstract of the paper do not reflect the high quality of work.

2.  The writing style is too weak like "Artificial jamming mainly in- 25 cludes two types. one is unintentional jamming [1]. The other is intentional jamming, 26 which refers to the jamming behavior taken for the purpose of destroying or disturbing 27 the information transmission process of a wireless communication system. " 

3. ACK is not defined 

4.  The sentence like " And this paper takes the probability of false alarm and missed detection into 89 account in anti-jamming communication for the first time, which fills th.." makes no sense to start it from the And.

5.  Figures 1 and 2 are not clear and no description is given that how the signal are jammed.

6.  Figure 7, 8  and 9 are very poor and not clear to the readers that what authors want to say here, how they addressed the issue and what will be the outcomes. 

7.  Some more work is needed on the antijaming technique with using modern mechanisms for evaluations the performance of proposed model. 

8. Furthermore, I failed to find the qualitative analysis in the conclusion and abstract. 

Author Response

Response to reviewer 4.

Round 2

Reviewer 1 Report

The authors have addressed almost my concerns. However, there are still minor comments to the authors:

1. The authors should submit a clear version without corrected parts. The authors can highlight changes on the PDF file, instead of corrected parts.

2. “Figure 5. Linear sweep jamming distribution” should be placed at the same page with Figure 5.

3. There is a white space at page 10 of 13.

4. Quality of some figures should be enhanced.

Author Response

Thank you for your comments. Here is my reply to you.

Reviewer 2 Report

The document was deeply improved, including several corrections and additional references. The authors have successfully answered by concerns from the first version of the paper.

However, the authors should improve the document, by correcting the following aspects:

- The meaning of the acronym POMDP is missing.

- Transmission power of transmitter P should be preferably given in dBm, not in W.

- Use "Fig." instead of "Figure" within the text (e.g Figure 1 shows... -> Fig. 1 shows...)

- Use "equation" instead of "eq." within the text (e.g ...update the Q value as eq. 19. -> ...update the Q value as equation 19.)

- Use Roman numeral for Tables (e.g Table I instead of Table 1)

- Use "Table I" instead of "table I" within the text (e.g According to the parameters in table... -> According to the parameters in Table...).

Author Response

(The authors gave the same response as above.)

Reviewer 4 Report

The authors have addressed my comments 

Author Response

Thank you for your review!